# Impact of abolishing prescription fees in Scotland on hospital admissions and prescribed medicines: an interrupted time series evaluation

Andrew James Williams,[1] William Henley,[2] John Frank[3]

[1]European Centre for Environment and Human Health, College of Medicine and Health, University of Exeter, Truro, UK
[2]Institute of Health Research, College of Medicine and Health, University of Exeter, Exeter, UK
[3]Scottish Collaboration for Public Health Research and Policy, University of Edinburgh, Edinburgh, UK

**Correspondence to**
Professor John Frank;
john.frank@ed.ac.uk

## ABSTRACT

**Objectives** To identify whether the abolition of prescription fees in Scotland resulted in: (1) Increase in the number (cost to NHS) of medicines prescribed for which there had been a fee (inhaled corticosteroids). (2) Reduction in hospital admissions for conditions related to those medications for which there had been a fee (asthma or chronic obstructive pulmonary disease (COPD))—when both are compared with prescribed medicines and admissions for a condition (diabetes mellitus) for which prescriptions were historically free.

**Design** Natural experimental retrospective general practice level interrupted time series (ITS) analysis using administrative data.

**Setting** General practices, Scotland, UK.

**Participants** 732 (73.6%) general practices across Scotland with valid dispensed medicines and hospital admissions data during the study period (July 2005–December 2013).

**Intervention** Reduction in fees per dispensed item from April 2008 leading to the abolition of the fee in April 2011, resulting in universal free prescriptions.

**Primary and secondary outcomes** Hospital admissions recorded in the Scottish Morbidity Record – 01 Inpatient (SMR01) and dispensed medicines recorded in the Prescribing Information System (PIS).

**Results** The ITS analysis identified marked step reductions in adult (19–59 years) admissions related to asthma or COPD (the intervention group), compared with older or young people with the same conditions or adults with diabetes mellitus (the counterfactual groups). The prescription findings were less coherent and subsequent sensitivity analyses found that both the admissions and prescriptions data were highly variable above the annual or seasonal level, limiting the ability to interpret the findings of the ITS analysis.

**Conclusions** This study did not find sufficient evidence that universal free prescriptions was a demonstrably effective or ineffective policy, in terms of reducing hospital admissions or reducing socioeconomic inequality in hospital admissions, in the context of a universal, publicly administered medical care system, the National Health Service of Scotland.

## Strengths and limitations of this study

► Administrative data permitted the undertaking of a large long-term evaluation of a policy, which otherwise would have been very costly and indeed, perhaps infeasible.
► Data were available for over 70% of the practices in Scotland, which could be used to classify those who did and did not experience a change in prescription fees.
► The data were only available at the general practice, not the individual level.
► The time series for both admissions and prescriptions were highly variable, preventing the drawing of specific conclusions from the evaluation.

## INTRODUCTION

When the National Health Service (NHS) was founded in the UK in 1948, all the services provided were free. However, as early as 1950 expenditure was higher than expected, so a charge (1 shilling) for prescribed items (including medicines, medical devices, dressings, borderline substances, etc) was introduced in 1952.[1] This prescription fee has increased over time and now in England the charge is £8.80 per item.[2] In 1968 a list of medical conditions for which no prescription fee would be charged was introduced; these were easily recognisable, lifelong, life-threatening conditions that required regular, prescribed medication.[1] Over time further exemptions were identified and the most recent list of exemption criteria is very extensive indeed (online supplementary file 1).[2] Now, the majority of prescriptions are exempt, meaning that in 2004/2005 less than 5% of the cost of pharmaceutical services in Scotland was covered by the prescription fee.[1]

The prescription fee has been described as a 'tax on ill health' (Nicola Sturgeon, then Scottish Health Secretary).[3 4] Although, the exemption criteria (online supplementary file 1) contain many chronic diseases, important conditions such as asthma and degenerative neurological conditions are missing. These omissions have led many to

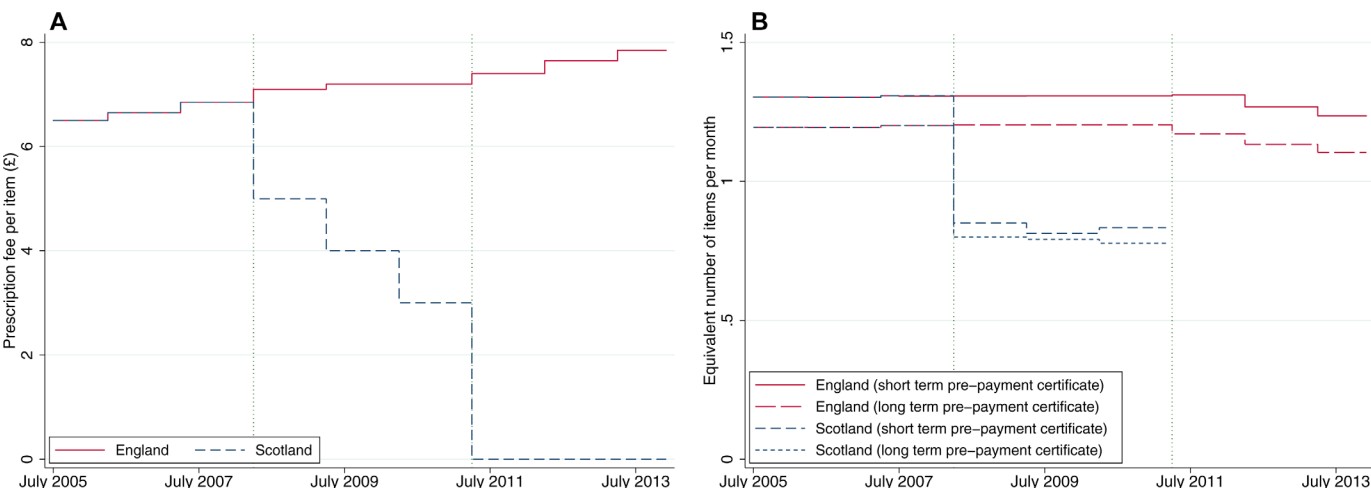

**Figure 1** Prescription fees in Scotland compared with England; (A) fee per item, (B) equivalent number of items purchased per month using prepayment certificates.

call for the abolition of the fee.[1 5 6] In 2007 Wales was the first country in the UK to abolish prescription fees, followed by Northern Ireland in 2010 and Scotland in 2011; a fee is still charged in England.[3] Figure 1A shows prescription fees in England and Scotland during the time period examined in this study. Patients requiring multiple prescriptions a month could purchase a short-term (3–4 month) or long-term (12 month) prepayment certificate (PPC) which provided a discount. Alongside the prescription fee reduction, Scotland made a marked reduction in PPC cost (figure 1B).

Although, the prescription fee only covers a small proportion of the costs of prescription services, abolishing the fee means that the cost needs to be met elsewhere. One of the justifications for abolishing prescription fees has been that this will lead to improved adherence to medication and subsequently reduced medical costs, through less need for expensive secondary medical care. Campbell *et al*[7] found that for patients with asthma in USA, an increase as small as $5 in pharmacy co-payments resulted in reduced adherence and increased outpatient visits. Kulik *et al*[8] undertook a randomised comparison in USA of the health impact of providing patients with recent myocardial infarction with free prescriptions for all medications, versus only a small selection of cardiovascular medications. It was found that full prescription cost coverage resulted in significantly improved adherence and trends towards reductions in rates of major vascular events or revascularisation.[8] In Canada, where all prescriptions are free for those over 65 years of age, health inequalities by socioeconomic position among those with diabetes are much reduced in this age group compared with those under 65 years of age.[9] Thus far, the only published evaluations of the policy change across the UK have come from Wales and only included prescriptions data and anecdotal reports around other outcomes.[10–13] Hence, the intention of this study was to identify whether the abolition of prescription fees in Scotland resulted in (see online supplementary file 2 for the study protocol):

A. increase in the number (cost to NHS) of medicines prescribed for which there had been a fee (inhaled corticosteroids);
B. Reduction in hospital admissions for conditions related to those medications for which there had been a fee (asthma and Chronic Obstructive Pulmonary Disease (COPD)).

The comparisons for both prescriptions and admissions were conditions for which prescriptions were historically free (diabetes mellitus, or the young or elderly). Subsequent research questions were:
C. Was there any change in hospital admissions in the practices which also saw a change in dispensing?
D. How did the effects of fee abolition differ across the socioeconomic spectrum?

## METHODS
### Retrospective policy analysis
As the abolition of prescription fees had already taken place and was not implemented experimentally, it was necessary to use a natural experimental approach: interrupted time series (ITS). This required historical data on policy exposure and outcomes.

### Data
Two large administrative data sets are available in Scotland which made this study feasible; the Prescribing Information System (PIS) and the Scottish Morbidity Record – 01 Inpatient (SMR01). PIS is the database of all community prescribed and dispensed medicines across Scotland. SMR01 is the database of all hospital inpatient admissions across Scotland. Using the unique identifier, the Community Health Index (CHI), it is possible to link individual health records across health databases in Scotland. However, CHI has only been recorded in PIS since August 2009, and therefore this study was restricted to general practice level data. The time period examined in this study was July 2005 to December 2013 with fee

reductions beginning in April 2008 and abolished from April 2011.

## Exposure

During study development, a group of clinicians and pharmacists was consulted to inform the study design, in particular the definition of exposure to the policy. Starting from the prescription fee exemption criteria (online supplementary file 1), groups for wom prescriptions remained free (counterfactual) and became free (intervention) needed defining. The groups needed to be identifiable in PIS and SMR01 and relate to relatively common conditions for which non-adherence to medication would likely result in the need for hospital admission. Subsequently, the subset of pharmaceutical-sensitive ambulatory care-sensitive conditions was used.[14] No data on patient financial situation is recorded in SMR01, which meant that intervention and counterfactual groups could not be defined in terms of the finance-related prescription fee exemption criteria. Research question (D) was included so that some examination of the impact of financial situation was undertaken. Asthma and COPD, common adulthood conditions with condition-specific mediations for which the prescription fee was charged, were identified as the most appropriate intervention conditions. Diabetes mellitus, medication for which had been exempt from the prescription fee for many years, constituted the counterfactual condition. In addition to a condition-based counterfactual group, an age-based counterfactual group was used. The criteria for the intervention and two counterfactual groups are given in table 1. The need to use the exemption criteria to classify intervention and counterfactuals meant that dispensing data from PIS was used, as exemption is only recorded when the prescription is dispensed. This prevented the study's consideration of primary non-adherence, when the patient does not have the prescription dispensed.

Using the exemption criteria to define the intervention and counterfactual groups within the PIS presented a challenge, as the exemption criteria did not need to be recorded after the fee abolition. Fortuitously, exemption criteria continued to be recorded in many instances. The proportion of prescriptions against each exemption criteria was compared before and after the abolition, to check if reporting had changed. Apart from those receiving free prescriptions due to being aged 60 years or over, reporting of exemption criteria fell markedly, particularly reporting PPCs or full payment. Consequently, given that Scottish Public Health Observatory data[15–17] did not show any marked changes in the prevalence of asthma, COPD or diabetes during the study period, the proportion eligible for each exemption prior to the policy change was applied to the PIS data after the policy change. This was not an issue for the SMR-01 data, as the variables used to define the groups (table 1) were not affected by the policy change. The data sets were prepared, cleaned and merged in Stata[18] with the top 1% of prescription data being deleted as these very large values appeared to be errors (most likely due to keying errors during manual data entry). To make the outcomes comparable between groups, the primary metric of prescriptions used was defined daily dose (DDDs). The prescription cost was also examined in terms of the gross ingredient cost before any discounts (dispensing costs are met by an overall service remuneration model within the NHS in Scotland rather than an additional cost per item). Number and quantity of prescriptions were provided in the data set but were found not to be comparable across study groups, so were not analysed. Data on the following characteristics of each general practice was also collected: health board, number of general practitioners, list size (by age group), whether the practice had its own dispensary, the practice NHS contract type, the quintile of the Scottish Index of Multiple Deprivation (SIMD)[19] of the practice postcode, the urban rural classification of the practice postcode, the number of patients registered at the practice living in a postcode defined as the 15% most deprived in Scotland, and the number of registered patients from each SIMD quintile and urban rural classification.

| Table 1 | Intervention and counterfactual group definitions | |
| --- | --- | --- |
| | **Prescribing Information System (PIS)** | **Scottish Morbidity Record – 01 Inpatient (SMR01)** |
| Intervention | BNF: 3.2 Inhaled steroids<br>Fee: paid<br>Age group: >18 years to <60 years | ICD-10: J20, J40X, J41, J42, J43, J44, J45, J46 or J47<br>Age group: >18 years to <60 years |
| Condition-based counterfactual | BNF: 6.1.1 or 6.1.2<br>Fee: exempt E—medical exemption<br>Age group: >18 years to <60 years | ICD-10: E10, E11, E12, E13 or E14<br>Age group: >18 years to <60 years |
| Age-based counterfactual | BNF: 3.2 Inhaled steroids<br>Fee: exempt A: <16 years, B: ≥16 years to ≤18 years and in full time education or C: ≥60 years<br>Age group: <16 years, ≥16 years to ≤18 years or ≥60 years | ICD-10: J20, J40X, J41, J42, J43, J44, J45, J46 or J47<br>Age group: <16 years, ≥16 years to ≤18 years or ≥60 years |

BNF, British National Formulary; ICD-10, International Classification of Disease 10th version.

## ITS analysis

The data set was declared a time series data set with monthly time periods and any missing time points within practices were filled with zero. As these data are related to instances when someone of a specific age with a specific condition from a specific general practice had been admitted to hospital or dispensed a medicine, missing data represent months when no prescriptions of that type or admissions for that condition occurred from that practice. Admissions included small and zero counts (eg, number of adult (19–59 years) asthma admissions per practice per month) so these data were analysed using mixed-effects Poisson models, offset by practice size to account for variations in practice size. DDDs and costs were transformed into DDDs or cost per 100 patients per practice to account for differences in practice size. The resulting measures were sufficiently normally distributed to be analysed using mixed-effects linear models. ITS operators were derived so that any model would fit a new intercept and slope at the time of the intervention, which could be delayed using lags. Calendar month was fitted as a categorical variable to account for seasonality. Finally, the data were treated as hierarchical with measurement month (i) nested within general practice (j) and the linear model for costs and DDDs was given by:

$$y_{ij} = \beta_{0ij} + \beta_1 date_{ij} + \beta_2 jan_{ij} + \beta_3 feb_{ij} + \beta_4 mar_{ij} + \beta_5 may_{ij}$$
$$+ \beta_6 jun_{ij} + \beta_7 jul_{ij} + \beta_8 aug_{ij} + \beta_9 sep_{ij} + \beta_{10} oct_{ij}$$
$$+ \beta_{11} nov_{ij} + \beta_{12} dec_{ij} + \beta_{13} reduction\ period_{ij}$$
$$+ \beta_{14} time\ since\ reductions\ began_{ij} + \beta_{15} abolition\ period_{ij}$$
$$+ \beta_{16} time\ since\ abolition\ began_{ij} + e_j + \mu_{ij}$$

Where $y_{ij}$ is the outcome for practice $j$ at time $i$ (in months), $date_{ij}$ is the count of months from the beginning of the study period (July 2005), $e_j$ is the difference between the intercept for the estimated overall trend and the intercept for the estimated trend for practice $j$ and $\mu_{ij}$ is the difference between the observed outcome value for time $i$ at practice $j$ and the predicted value, and $e_j$ and $\mu_{ij}$ are random effects. $Jan_{ij}$ to $dec_{ij}$ are dummy variables for each month from January to December, with April being the reference category as this was the month in which the policy changes were implemented. $Reduction\ period_{ij}$ and $abolition\ period_{ij}$ are dummy variables for the periods April 2008 to March 2011, and April 2011 to December 2013 (the end of the study period), respectively. $Time\ since\ reductions\ began_{ij}$ and $time\ since\ abolition\ began_{ij}$ are counts of months from the beginning to end of the reduction and abolition periods, respectively. The meaning of the β coefficients were: $\beta_0$ mean outcome in July 2005 (intercept), $\beta_1$ secular trend for time from July 2005, $\beta_{2-12}$ mean effect for each month excluding April (seasonality), $\beta_{13}$ change in intercept when the prescription fee was being reduced (reduction step change), $\beta_{14}$ change in slope (from April 2008 to March 2011) when the prescription fee was being reduced (reduction change in slope), $\beta_{15}$ change in intercept when the prescription fee was abolished (abolition step change) and $\beta_{16}$ change in slope (from April 2011) when the prescription fee was abolished (abolition

change in slope). The Poisson model for admissions was defined similarly with the logarithm of the admission rate expressed as a linear function of the fixed and random effects, and with the addition of an offset term for the practice size.

Admissions data were analysed with 0-month, 1-month, 2-month and 3-month lags on the intervention effect. To address research question (C), it was planned that practices where changes in prescriptions and admissions consistent with the hypothesised effect of the policy change be identified and compared as a secondary analysis. Finally, to address research question (D), practices whose population modal quintile of the SIMD 2012[19] remained the same throughout the study period (constant sample size) were included in a repetition of the primary analysis stratified by quintile of SIMD. During model development the hierarchical linear time series models were specified with first-order or second-order autoregression or moving averages. The results did not differ substantially between specifications and therefore all the results presented were specified with second-order moving average.

## Patient and public involvement

In the early phases of development (January 2015), this project was presented to the public panel of the Scottish Farr Institute (http://www.farrinstitute.org/farr-scotland). The ensuing discussion demonstrated the interest of the panel members in the project, which is critical for secondary data projects such as this, and helped shape the analysis and interpretation of the findings. In particular, the panel members brought to the researchers' attention the issue of stockpiling of prescribed medicines following the fee abolition. As a secondary data analysis there was limited potential for public engagement during the conduct of the study, but the results will be disseminated to the public panel and more widely through the Farr Institute website.

## RESULTS

Across the 8-year study period, the number of practices and their characteristics changed. The characteristics of the practices with prescriptions data at the end of the study period (December 2013) are presented in table 2. Data on prescribed medicines were available for 732 (73.6%) of the 994 general practices across Scotland at the end of 2013, while admissions data were available for 754 (75.9%) practices. The majority of practices had five or fewer general practitioners, fewer than 10 000 patients and were in urban areas. Less than 10% of practices had their own dispensary.

The intercept (constant) and coefficients of the ITS operators for the Poisson and linear mixed-effects models of admissions, DDDs and cost are presented in table 3. Throughout the results, p values have not been reported and the focus was on estimated effect sizes, as the size of the data set resulted in many statistical but not meaningful significances. The intercept estimates demonstrate

**Table 2** Practice characteristics in December 2013 (n=732)

| Characteristic | Summary statistic |
|---|---|
| Number of GPs | |
| ≤5 general practitioners | 447 (61.1%) |
| >5–≤10 general practitioners | 246 (33.6%) |
| >10 general practitioners | 39 (5.3%) |
| Practice size | |
| ≤5000 patients | 345 (47.1%) |
| >5000–≤10 000 patients | 305 (41.7%) |
| >10 000 patients | 82 (11.2%) |
| Dispensing practice | 67 (9.2%) |
| Health board | |
| NHS Ayrshire and Arran | 37 (5.1%) |
| NHS Borders | 20 (2.7%) |
| NHS Dumfries and Galloway | 26 (3.6%) |
| NHS Fife | 40 (5.5%) |
| NHS Forth Valley | 44 (6.0%) |
| NHS Grampian | 37 (5.1%) |
| NHS Greater Glasgow and Clyde | 215 (29.4%) |
| NHS Highland | 72 (9.8%) |
| NHS Lanarkshire | 60 (8.2%) |
| NHS Lothian | 107 (14.6%) |
| NHS Orkney | 3 (0.4%) |
| NHS Shetland | 4 (0.5%) |
| NHS Tayside | 60 (8.2%) |
| NHS Western Isles | 7 (1.0%) |
| Modal SIMD quintile of patients | |
| Quintile 1 (most deprived) | 224 (30.6%) |
| Quintile 2 | 127 (17.3%) |
| Quintile 3 | 155 (21.2%) |
| Quintile 4 | 108 (14.8%) |
| Quintile 5 (least deprived) | 118 (16.1%) |
| Modal location of patients | |
| Large urban areas | 302 (41.3%) |
| Other urban areas | 192 (26.2%) |
| Accessible small towns | 64 (8.7%) |
| Remote small towns | 17 (2.3%) |
| Very remote small towns | 9 (1.2%) |
| Accessible rural | 61 (8.3%) |
| Remote rural | 32 (4.4%) |
| Very remote rural | 55 (7.5%) |

GP, general practitioner; NHS, National Health Service; SIMD, Scottish Index of Multiple Deprivation.[19]

that there were pre-existing differences in the level of admissions and prescriptions across the intervention and counterfactual groups at the start of the study period (July 2005). The highest rate of admissions, and number (DDDs) and cost of prescribed medicines were observed in the age-based counterfactual group, while the fewest admissions and prescribed medicines were seen among the intervention group. This demonstrates that asthma and COPD are, as expected, most problematic to the young and old. Despite these differences in quantity, the background secular time trend in admissions before the abolition of prescription fees is consistent across the groups. The coefficients of the ITS operators are also mostly consistent across groups and increasing degrees of time lag. A notable exception was a significant step reduction in admissions in the intervention group, related to the abolition of prescription fees, although the underlying time trend did not significantly change. The results regarding prescribed medicines are much less clear, with more marked temporal variability across the study period. However, the largest changes appear in the counterfactual groups, indicating some phenomenon other than the intended effect of the policy change, which we were unable to identify.

The results of the analysis undertaken to examine whether the impact of the policy differed by socioeconomic position are presented in table 4. These analyses aimed to assess whether the financial exemptions to the prescription fee (online supplementary file 1) had an impact on the results, as these exemptions could not be accounted for directly in the main analysis. Only the results for the intervention group have been presented, as individual financial position should not have had an impact on the counterfactual groups. The intercepts of the admissions analyses are consistent with known inequalities, with the highest rate of admissions among the most deprived (Q1), and the lowest rate of admissions in the least deprived (Q5). The results for the ITS operators are similar to the main analysis, with the most marked differences being observed in terms of step changes as opposed to changes in the underlying time trend. Although not statistically significant, there seems to be some evidence of reductions in admissions among the most deprived quintiles (Q1 and Q2) in relation to the fee-reduction policy, compared with the least deprived quintiles (Q4 and Q5). However, any effect is not linear across the SIMD quintiles. Again, the results of the analysis of prescriptions were less coherent. The intercepts of the models demonstrate that most paid prescriptions were from practices with mostly Q3 and Q4 patients, who are likely to have had poorer health than Q5 patients, but would be less likely to be eligible for free prescriptions for financial reasons than Q1 and Q2 patients. These quintiles also demonstrate the largest step increases in DDDs when the fees were abolished, which is consistent with this being an effect of the policy.

In response to the incoherence of the findings for DDDs and costs and reviewer comments, several sensitivity analyses were undertaken, instead of the analyses originally planned to address the third research question (C). First, using the generalised additive mixed models (GAMM) package in R,[20–24] the models were fitted without the ITS operators, but by fitting eight knots evenly distributed

**Table 3** Results of interrupted time series analysis of the impact of reducing and abolishing prescription fees in Scotland on hospital admissions (incidence rate ratios) and prescribed medicines (defined daily doses and cost)

| | Intercept | | Monthly change | | Reduction step change | | Reduction change in slope | | Abolition step change | | Abolition change in slope | |
|---|---|---|---|---|---|---|---|---|---|---|---|---|
| **Admissions** | IR | 95% CI | IRR | 95% CI | IRR | 95% CI | IRR | 95% CI | IRR | 95% CI | IRR | 95% CI |
| Intervention | $1.01e^{-2}$ | $9.46e^{-3}$ to $1.09e^{-2}$ | 1.00 | 1.00 to 1.01 | 1.03 | 0.98 to 1.08 | 0.99 | 0.99 to 0.99 | 0.81 | 0.72 to 0.90 | 1.00 | 0.99 to 1.00 |
| Age-c'fact | $6.65e^{-2}$ | $6.36e^{-2}$ to $6.95e^{-2}$ | 1.00 | 1.00 to 1.00 | 1.03 | 1.01 to 1.06 | 1.00 | 0.99 to 1.00 | 0.95 | 0.90 to 1.00 | 1.00 | 0.99 to 1.00 |
| Condition-c'fact | $4.80e^{-3}$ | $4.39e^{-3}$ to $5.24e^{-3}$ | 1.00 | 1.00 to 1.01 | 0.95 | 0.88 to 1.03 | 1.00 | 1.00 to 1.01 | 1.01 | 0.86 to 1.20 | 1.00 | 1.00 to 1.01 |
| **Admissions 1-month lag** | IR | 95% CI | IRR | 95% CI | IRR | 95% CI | IRR | 95% CI | IRR | 95% CI | IRR | 95% CI |
| Intervention | $1.02e^{-2}$ | $9.49e^{-3}$ to $1.09e^{-2}$ | 1.01 | 1.00 to 1.01 | 1.01 | 0.96 to 1.06 | 0.99 | 0.99 to 0.99 | 0.78 | 0.71 to 0.87 | 1.00 | 0.99 to 1.00 |
| Age-c'fact | $6.74e^{-2}$ | $6.45e^{-2}$ to $7.05e^{-2}$ | 1.00 | 1.00 to 1.00 | 1.03 | 1.01 to 1.06 | 1.00 | 0.99 to 1.00 | 0.95 | 0.90 to 1.00 | 1.00 | 0.99 to 1.00 |
| Condition-c'fact | $4.83e^{-3}$ | $4.41e^{-3}$ to $5.28e^{-3}$ | 1.00 | 1.00 to 1.00 | 0.98 | 0.91 to 1.06 | 1.00 | 1.00 to 1.00 | 1.07 | 0.91 to 1.26 | 1.00 | 1.00 to 1.01 |
| **Admissions 2-month lag** | IR | 95% CI | IRR | 95% CI | IRR | 95% CI | IRR | 95% CI | IRR | 95% CI | IRR | 95% CI |
| Intervention | $1.02e^{-2}$ | $9.51e^{-3}$ to $1.09e^{-2}$ | 1.00 | 1.00 to 1.01 | 1.01 | 0.96 to 1.06 | 0.99 | 0.99 to 0.99 | 0.79 | 0.71 to 0.87 | 1.00 | 0.99 to 1.00 |
| Age-c'fact | $6.77e^{-2}$ | $6.47e^{-2}$ to $7.07e^{-2}$ | 1.00 | 1.00 to 1.00 | 1.04 | 1.02 to 1.07 | 1.00 | 0.99 to 1.00 | 0.97 | 0.92 to 1.02 | 1.00 | 1.00 to 1.00 |
| Condition-c'fact | $4.85e^{-3}$ | $4.44e^{-3}$ to $5.30e^{-3}$ | 1.00 | 1.00 to 1.00 | 0.99 | 0.92 to 1.07 | 1.00 | 1.00 to 1.01 | 1.10 | 0.93 to 1.28 | 1.00 | 1.00 to 1.01 |
| **Admissions 3-month lag** | IR | 95% CI | IRR | 95% CI | IRR | 95% CI | IRR | 95% CI | IRR | 95% CI | IRR | 95% CI |
| Intervention | $1.02e^{-2}$ | $9.49e^{-2}$ to $1.09e^{-2}$ | 1.01 | 1.00 to 1.01 | 0.99 | 0.94 to 1.04 | 0.99 | 0.99 to 0.99 | 0.78 | 0.70 to 0.86 | 1.00 | 0.99 to 1.00 |
| Age-c'fact | $6.73e^{-2}$ | $6.44e^{-2}$ to $7.04e^{-2}$ | 1.00 | 1.00 to 1.00 | 1.02 | 1.00 to 1.05 | 1.00 | 0.99 to 1.00 | 0.94 | 0.90 to 0.99 | 1.00 | 1.00 to 1.00 |
| Condition-c'fat | $4.82e^{-3}$ | $4.41e^{-3}$ to $5.26e^{-3}$ | 1.00 | 1.00 to 1.00 | 0.97 | 0.90 to 1.05 | 1.00 | 1.00 to 1.01 | 1.08 | 0.92 to 1.26 | 1.00 | 1.00 to 1.00 |
| **DDDs** | DDDs | 95% CI | ΔDDDs | 95% CI | ΔDDDs | 95% CI | ΔDDDs | 95% CI | ΔDDDs | 95% CI | ΔDDDs | 95% CI |
| Intervention | 29.65 | 28.71 to 30.59 | −0.04 | −0.06 to −0.02 | −0.11 | −0.61 to 0.39 | 0.04 | 0.02 to 0.07 | 6.03 | 4.94 to 7.13 | 0.10 | 0.08 to 0.13 |
| Age-c'fact | 100.88 | 98.07 to 103.70 | −0.05 | −0.11 to 0.01 | 5.06 | 3.61 to 6.52 | 0.68 | 0.59 to 0.77 | 38.37 | 34.84 to 41.90 | 0.34 | 0.25 to 0.42 |
| Condition-c'fact | 55.61 | 53.70 to 57.53 | 0.14 | 0.10 to 0.18 | −1.88 | −2.90 to −0.85 | −0.05 | −0.11 to 0.00 | 6.02 | 3.73 to 8.32 | 0.15 | 0.09 to 0.20 |
| **Cost** | £s | 95% CI | Δ£s | 95% CI | Δ£s | 95% CI | Δ£s | 95% CI | Δ£s | 95% CI | Δ£s | 95% CI |
| Intervention | 19.36 | 18.53 to 20.19 | 0.16 | 0.15 to 0.18 | 0.23 | −0.19 to 0.66 | −0.15 | −0.17 to −0.13 | −0.96 | −1.92 to −0.01 | −0.10 | −0.12 to −0.08 |
| Age-c'fact | 69.92 | 67.35 to 72.49 | 0.62 | 0.56 to 0.68 | 4.74 | 3.42 to 6.06 | 0.03 | −0.05 to 0.12 | 14.24 | 10.86 to 17.63 | −0.36 | −0.44 to −0.28 |
| Condition-c'fact | 26.35 | 25.35 to 27.34 | 0.15 | 0.13 to 0.17 | −3.16 | −3.70 to −2.62 | −0.01 | −0.04 to 0.02 | 2.00 | 0.79 to 3.20 | −0.11 | −0.14 to −0.08 |

c'fact, counterfactual; DDDs, defined daily doses per 100 patients per month; ΔDDDs, change in DDDs; IR, incidence rate per 100 patients per practice per month; IRR, incidence rate ratio; £s, gross ingredient cost of medicines before any discount per 100 patients per practice per month; Δ£s, change in £s.

**Table 4** Results of interrupted time series analysis of the impact of reducing and abolishing prescription fees in Scotland on hospital admissions (incidence rate ratios) and prescribed medicines (defined daily doses and cost) stratified by quintile of the Scottish Index of Multiple Deprivation[19]

| | Intercept | | Monthly change | | Reduction step change | | Reduction change in slope | | Abolition step change | | Abolition change in slope | |
|---|---|---|---|---|---|---|---|---|---|---|---|---|
| **Admissions** | IR | 95% CI | IRR | 95% CI | IRR | 95% CI | IRR | 95% CI | IRR | 95% CI | IRR | 95% CI |
| Q1—most deprived | $1.52e^{-2}$ | $1.35e^{-2}$ to $1.71e^{-2}$ | 1.01 | 1.01 to 1.01 | 1.00 | 0.91 to 1.09 | 0.99 | 0.98 to 0.99 | 0.64 | 0.52 to 0.78 | 0.99 | 0.98 to 0.99 |
| Q2 | $1.17e^{-2}$ | $9.36e^{-3}$ to $1.47e^{-2}$ | 1.00 | 1.00 to 1.01 | 1.10 | 0.92 to 1.32 | 0.99 | 0.98 to 1.00 | 0.75 | 0.50 to 1.12 | 1.00 | 0.99 to 1.01 |
| | $9.99e^{-3}$ | $7.45e^{-3}$ to $1.34e^{-2}$ | 1.00 | 0.99 to 1.01 | 1.10 | 0.86 to 1.41 | 1.00 | 0.98 to 1.01 | 1.15 | 0.68 to 1.94 | 1.00 | 0.99 to 1.02 |
| Q4 | $6.49e^{-3}$ | $4.64e^{-3}$ to $9.07e^{-3}$ | 0.99 | 0.98 to 1.00 | 1.50 | 1.13 to 1.99 | 1.00 | 0.99 to 1.02 | 1.63 | 0.89 to 2.99 | 1.02 | 1.01 to 1.04 |
| Q5—least deprived | $6.06e^{-3}$ | $4.86e^{-3}$ to $7.54e^{-3}$ | 1.00 | 0.99 to 1.00 | 1.00 | 0.84 to 1.21 | 1.00 | 1.00 to 1.01 | 1.24 | 0.85 to 1.82 | 1.00 | 0.99 to 1.01 |
| **Admissions 1-month lag** | IR | 95% CI | IRR | 95% CI | IRR | 95% CI | IRR | 95% CI | IRR | 95% CI | IRR | 95% CI |
| Q1—most deprived | $1.50e^{-2}$ | $1.33e^{-2}$ to $1.69e^{-2}$ | 1.01 | 1.01 to 1.02 | 0.94 | 0.86 to 1.03 | 0.99 | 0.98 to 0.99 | 0.60 | 0.49 to 0.73 | 0.99 | 0.98 to 0.99 |
| Q2 | $1.20e^{-2}$ | $9.58e^{-3}$ to $1.51e^{-2}$ | 1.00 | 1.00 to 1.01 | 1.12 | 0.93 to 1.34 | 0.99 | 0.98 to 1.00 | 0.79 | 0.53 to 1.16 | 1.00 | 0.99 to 1.01 |
| | $9.97e^{-3}$ | $7.40e^{-3}$ to $1.34e^{-2}$ | 1.00 | 0.99 to 1.01 | 1.00 | 0.78 to 1.27 | 1.00 | 0.99 to 1.01 | 0.99 | 0.60 to 1.65 | 1.00 | 0.99 to 1.02 |
| Q4 | $6.81e^{-3}$ | $4.85e^{-3}$ to $9.56e^{-3}$ | 0.99 | 0.98 to 1.00 | 1.53 | 1.16 to 2.02 | 1.00 | 0.99 to 1.01 | 1.66 | 0.91 to 3.00 | 1.02 | 1.01 to 1.04 |
| Q5—least deprived | $6.12e^{-3}$ | $4.90e^{-3}$ to $7.64e^{-3}$ | 1.00 | 0.99 to 1.00 | 1.03 | 0.86 to 1.23 | 1.01 | 1.00 to 1.01 | 1.27 | 0.87 to 1.85 | 1.00 | 0.99 to 1.01 |
| **Admissions 2-month lag** | IR | 95% CI | IRR | 95% CI | IRR | 95% CI | IRR | 95% CI | IRR | 95% CI | IRR | 95% CI |
| Q1—most deprived | $1.51e^{-2}$ | $1.34e^{-2}$ to $1.70e^{-2}$ | 1.01 | 1.01 to 1.01 | 0.94 | 0.86 to 1.03 | 0.99 | 0.98 to 0.99 | 0.60 | 0.50 to 0.73 | 0.99 | 0.98 to 0.99 |
| Q2 | $1.20e^{-2}$ | $9.58e^{-3}$ to $1.51e^{-2}$ | 1.00 | 1.00 to 1.01 | 1.11 | 0.92 to 1.33 | 0.99 | 0.98 to 1.00 | 0.78 | 0.54 to 1.15 | 1.00 | 0.99 to 1.01 |
| | $1.02e^{-2}$ | $7.59e^{-3}$ to $1.37e^{-2}$ | 1.00 | 0.99 to 1.01 | 1.07 | 0.84 to 1.37 | 1.00 | 0.99 to 1.01 | 1.12 | 0.68 to 1.84 | 1.00 | 0.99 to 1.02 |
| Q4 | $6.77e^{-3}$ | $4.83e^{-3}$ to $9.48e^{-3}$ | 0.99 | 0.98 to 1.00 | 1.57 | 1.19 to 2.07 | 1.00 | 0.98 to 1.01 | 1.70 | 0.95 to 3.05 | 1.02 | 1.00 to 1.03 |
| Q5—least deprived | $6.21e^{-3}$ | $4.98e^{-3}$ to $7.75e^{-3}$ | 0.99 | 0.99 to 1.00 | 1.09 | 0.91 to 1.31 | 1.00 | 1.00 to 1.01 | 1.38 | 0.95 to 1.99 | 1.00 | 0.99 to 1.01 |
| **Admissions 3-month lag** | IR | 95% CI | IRR | 95% CI | IRR | 95% CI | IRR | 95% CI | IRR | 95% CI | IRR | 95% CI |
| Q1—most deprived | $1.52e^{-2}$ | $1.35e^{-3}$ to $1.71e^{-2}$ | 1.01 | 1.01 to 1.01 | 0.94 | 0.86 to 1.02 | 0.99 | 0.98 to 0.99 | 0.60 | 0.50 to 0.73 | 0.99 | 0.98 to 0.99 |
| Q2 | $1.18e^{-2}$ | $9.40e^{-3}$ to $1.48e^{-2}$ | 1.01 | 1.00 to 1.01 | 1.03 | 0.86 to 1.24 | 0.99 | 0.98 to 1.00 | 0.72 | 0.50 to 1.05 | 1.00 | 0.99 to 1.01 |
| | $1.05e^{-2}$ | $7.82e^{-3}$ to $1.41e^{-2}$ | 1.00 | 0.99 to 1.00 | 1.18 | 0.92 to 1.51 | 1.00 | 0.99 to 1.01 | 1.25 | 0.76 to 2.05 | 1.01 | 0.99 to 1.02 |
| Q4 | $6.40e^{-3}$ | $4.58e^{-3}$ to $8.96e^{-3}$ | 0.99 | 0.98 to 1.00 | 1.34 | 1.02 to 1.77 | 1.00 | 0.98 to 1.01 | 1.36 | 0.77 to 2.39 | 1.01 | 1.00 to 1.03 |
| Q5—least deprived | $6.15e^{-3}$ | $4.93e^{-3}$ to $7.67e^{-3}$ | 1.00 | 0.99 to 1.00 | 1.07 | 0.89 to 1.28 | 1.00 | 0.99 to 1.01 | 1.35 | 0.94 to 1.94 | 1.00 | 0.99 to 1.01 |
| **DDDs** | DDDs | 95% CI | ΔDDDs | 95% CI | ΔDDDs | 95% CI | ΔDDDs | 95% CI | ΔDDDs | 95% CI | ΔDDDs | 95% CI |
| Q1—most deprived | 21.56 | 20.13 to 22.99 | −0.07 | −0.10 to −0.04 | 0.25 | −0.50 to 1.01 | 0.08 | 0.04 to 0.12 | 6.23 | 4.59 to 7.86 | 0.14 | 0.10 to 0.18 |
| Q2 | 29.47 | 25.42 to 33.51 | 0.05 | −0.03 to 0.12 | −1.83 | −3.80 to 0.15 | −0.05 | −0.15 to 0.05 | 2.09 | −2.19 to 6.37 | 0.03 | −0.07 to 0.14 |
| Q3 | 35.64 | 31.29 to 39.98 | −0.06 | −0.18 to 0.06 | 1.11 | −2.07 to 4.28 | −0.03 | −0.20 to 0.13 | 7.32 | 0.27 to 14.37 | 0.15 | −0.02 to 0.32 |
| Q4 | 37.89 | 34.33 to 41.46 | −0.05 | −0.14 to 0.03 | 0.12 | −2.06 to 2.31 | 0.06 | −0.05 to 0.17 | 7.62 | 2.87 to 12.38 | 0.09 | −0.03 to 0.21 |
| Q5—least deprived | 30.20 | 28.03 to 32.37 | 0.02 | −0.02 to 0.06 | −1.15 | −2.24 to −0.05 | $9.05e^{-4}$ | −0.06 to 0.06 | 4.21 | 1.83 to 6.60 | 0.01 | −0.05 to 0.06 |
| **Cost** | £s | 95% CI | Δ£s | 95% CI | Δ£s | 95% CI | Δ£s | 95% CI | Δ£s | 95% CI | Δ£s | 95% CI |
| Q1—most deprived | 14.25 | 12.94 to 15.55 | 0.07 | 0.05 to 0.09 | 0.37 | −0.24 to 0.98 | −0.06 | −0.09 to −0.03 | 1.02 | −0.32 to 2.36 | −0.02 | −0.05 to 0.02 |
| Q2 | 21.61 | 17.63 to 25.58 | 0.21 | 0.14 to 0.28 | −0.88 | −2.67 to 0.90 | −0.21 | −0.30 to −0.12 | −2.63 | −6.59 to 1.32 | −0.18 | −0.28 to −0.09 |
| Q3 | 22.97 | 19.33 to 26.60 | 0.20 | 0.10 to 0.29 | 0.83 | −1.55 to 3.20 | −0.25 | −0.38 to −0.13 | −1.67 | −6.95 to 3.62 | −0.08 | −0.21 to 0.04 |

Continued

**Table 4** Continued

| | Intercept | | Monthly change | | Reduction step change | | Reduction change in slope | | Abolition step change | | Abolition change in slope | |
|---|---|---|---|---|---|---|---|---|---|---|---|---|
| Q4 | 24.84 | 21.53 to 28.15 | 0.22 | 0.14 to 0.30 | 0.14 | −1.92 to 2.21 | −0.24 | −0.35 to −0.13 | −2.63 | −7.26 to 2.01 | −0.19 | −0.30 to −0.08 |
| Q5—least deprived | 19.78 | 17.59 to 21.97 | 0.20 | 0.17 to 0.24 | −0.78 | −1.72 to 0.16 | −0.18 | −0.23 to −0.14 | −2.67 | −4.76 to −0.58 | −0.18 | −0.23 to −0.13 |

DDDs, defined daily doses per 100 patients per practice per month; ΔDDDs, change in DDDs; IR, incidence rate per 100 patients per practice per month; IRR, incidence rate ratio; £s, gross ingredient cost of medicines before any discount per 100 patients per practice per month; Δ£s, change in £s; Q1–5, quintiles of the Scottish Index of Multiple Deprivation from the most to the least deprived.

across the time series. These models allow change points in the time series at times other than those related to the policy to be revealed and relax the assumption that any time effects were linear. Plots of each of the models undertaken as part of the sensitivity analyses are shown in online supplementary file 3, with the dates of reduction and abolition overlaid. The admissions' data for the condition-based counterfactual group show that following the adjustment for seasonality there was a small but linear increase in diabetes-related admissions among adults, which, as expected, appears not to change in relation to the prescription fee policy changes. However, the admissions' data for the intervention-based and age-based counterfactual groups show marked changes which are neither consistent year to year, nor easily related to the policy changes. The patterns of admissions in both the intervention-based and age-based counterfactual groups are similar. Both the DDDs and cost of prescribed medicines data for all three groups show similar patterns of increases, with occasional change points. These change points appear to be spaced in time similarly to the reduction and abolition of prescription fees, but around 12 months after the policy changes. The magnitude of the changes in prescribed DDDs and costs are also greater in the counterfactuals than the intervention group, arguing against any causal effect of the intervention.

The second sensitivity analysis undertaken was to adjust for clustering by health board as well as general practice. The results of these models are presented in online supplementary file 4; apart from changes in the intercepts, the estimated coefficients did not change. As a third sensitivity analysis, in light of the results of the GAMM models, the ITS models were run with either a quadratic secular time effect added, or quadratic effects for time since reduction and time since abolition. The results of these models are presented in online supplementary file 5 alongside plots comparing the linear and quadratic effects. The coefficients of the quadratic time effects were minimal and resulted in very few marked changes in the other estimated coefficients. The second and third sensitivity analyses demonstrate that the findings of the ITS models were robust to some changes in the clustering of the data and time effects. However, the unusual trends in the time series identified within the first sensitivity analysis and the inconsistency between the findings and the hypothesised effects prevent the drawing of firm conclusions from this study.

## DISCUSSION

Using Scottish administrative hospital admissions and prescribed medicines data it was possible to undertake an evaluation of the impact of a major policy change. The evaluation included over 70% of the general practices in Scotland and analysed changes over an 8-year period. ITS analysis offered a natural experimental approach for evaluating the impact of policy changes which occurred on specific dates. There was a marked step reduction in

adult asthma-related or COPD-related admissions around the time of the abolition of prescription fees (April 2011), compared with the counterfactual groups. This effect was consistent with known health inequalities and the prescription fee policy prior to the change. The data on prescribed medicines were less coherent, with marked changes in the intervention and counterfactual groups. Sensitivity analyses revealed that there was considerable variation in almost all the time series that related neither to policy change, nor to season or annual changes. It was therefore difficult to draw any firm conclusions about the impact of the policy change.

More rigorous and conclusive results would be generated by an experimental study. However, a prospective study of this size and duration would have been considerably more costly, and potentially impossible to fund. Policy changes like the one evaluated are also rarely introduced experimentally. The prescription fee exemption criteria, rather ironically, made this study possible. One challenge to this study was that the prescription fee exemption criteria did not need to be reported following the policy change, so the assumption was made that the proportions eligible for each exemption criteria would not vary markedly in the relatively short period being studied following the policy change. This assumption was supported by the finding that the proportion receiving free prescribed medicines due to older age remained the same, as these people, possibly through habit, continued to report their exemption. However, the variability in the prescription time series limited our ability to draw any firm conclusions regarding prescribed medicines. It is a weakness of the study that it was only possible to examine general practice-level, rather than individual-level data. This prevented the assessment of primary non-adherence and was the result of data not being collected which would permit individual linkage. This linkage is now possible, but not for the years before the policy change. Both SMR-01 and PIS are administrative data sets, collected for billing and reimbursement purposes, and therefore data which are most useful to research are often not collected.

Unlike previous studies on the impact of universal free prescriptions, it has not been possible to identify a clear impact of the policy change in Scotland. The other evaluations of prescription fees have been located in USA and Canada, where the health systems are quite different to the UK's NHS. Kulik et al[8] were able to undertake a randomised controlled trial offering universal free prescriptions to patients postmyocardial infarction, compared with free prescriptions only for medication for the myocardial infarction. The participants in that study had a well-defined need for prescriptions, whereas in the current study—although the intervention group had a long-term condition (asthma or COPD)—the need for regular medication among adults with these conditions will vary considerably between patients. Like the current study, both Campbell et al[7] and Booth et al[9] used administrative data permitting larger retrospective studies. The study by Campbell et al[7] is very similar to the current study

but examined changes in monthly co-payments. They contrasted the impact of increases of greater or less than $5, which is approximately half the cost of the current English prescription fee. However, this was an increase from the average monthly co-payment of around $17, which is considerably larger than the English prescription fee. Booth et al[9] examined the whole population with diabetes over 65 years of age, who are more likely to be on numerous medications than the adult (19–59 years of age with asthma or COPD) intervention group in this study and therefore would have faced higher prescription costs in a health system where their prescriptions were not free. Finally, the data analysed in this study showed that prior to the policy change less than 13% of prescriptions were paid in full or by PPC. Subsequently, considering the proportion of that 13% that are for acute conditions, such as bacterial infections, this study may have been looking for a needle in a haystack.

Overall, in this study no marked impact of the abolition of prescription fees in Scotland on prescribed medicines or hospital admissions has been identifiable. However, there were some signs of some impact, specifically a step reduction, within 1 month of abolition, in adult admissions for asthma or COPD. Although, the availability of administrative data made this study possible, it has also brought the biggest limitations to the study. As the data were primarily collected for administrative and financial purposes, it was understandably not optimal for this study, one of the most significant challenges to the use of big data in research.[25–27] Although the PIS data were particularly messy, even having dropped the top 1% of values, the central recording of community prescriptions in Scotland, unlike England, made this study possible. This study also encountered one of the other challenges to research using big data, 'over sensitivity of statistical hypothesis testing'.[25 28] These are at least two of the topics for further research in the field of big data. Based on this study, it is not possible to conclude whether universal free prescription is an effective policy in terms of improving health or reducing inequalities; however, neither does the study provide evidence that the policy is ineffective. There has been some concern that not charging for prescriptions could lead to overuse or stockpiling, which for many medications is risky, as well as a waste of limited resources. The increases in prescriptions in the counterfactual groups may support this hypothesis. Charging a reasonable fee per prescription may limit this; however, the size of the fee required to cover the administrative costs of the payment system may lead to the fee being high enough to inappropriately limit pharmaceutical use. Rigorous evaluation needs to accompany any health policy changes to inform our understanding of the most efficient health systems and services.

**Acknowledgements** The authors thank the other members of the Farr Institute @ Scotland Natural Experimental Approaches research group for their support; Corri Black (University of Aberdeen), Chris Dibben (University of Edinburgh), Ruth Jepson (University of Edinburgh), Anne Ludbrook (University of Aberdeen) and Jill Pell (University of Glasgow). The authors also thank Marion Bennie (University of

Strathclyde) for support to this project and for the help in convening the following clinicians and pharmacists to inform the definition of the intervention and control groups: Simon Hurding, Sean MacBride-Stewart and Margaret Ryan. Finally, the authors also thank Stuart McTaggart and Doug Kidd for their help in accessing and understanding the data.

**Collaborators** Corri Black; Chris Dibben; Ruth Jepson; Anne Ludbrook; Jill Pell; Marion Bennie; Simon Hurding; Sean MacBride-Stewart; Margaret Ryan; Stuart McTaggart; Doug Kidd.

**Contributors** AJW contributed to the conceptualisation of the project, undertook all the statistical analysis and drafted the paper. JF contributed to the conceptualisation of the project, supervised the analysis, reviewed and approved the paper. WH supported AJW with the statistical analysis, reviewed and approved the paper.

**Funding** This work was funded by the Farr Institute @ Scotland, which is funded by the following consortium: Arthritis Research UK, the British Heart Foundation, Cancer Research UK, the Economic and Social Research Council, the Engineering and Physical Sciences Research Council, the Medical Research Council, the National Institute of Health Research, the National Institute for Social Care and Health Research (Welsh Assembly Government), the Chief Scientist Office (Scottish Government Health Directorates), (MRC Grant No: MR/K007017/1). JF and AJW worked for the Scottish Collaboration for Public Health Research and Policy which is funded by the SCPHRP core grant from the Medical Research Council (Grant Number MR/K023209/1) and the Chief Scientist Office of Scotland. WH received support from the National Institute for Health Research (NIHR) Collaboration for Leadership in Applied Health Research and Care (CLAHRC) for the South West Peninsula. The funders played no role in the conceptualisation or realisation of the research and no role in the decision to submit it for publication.

**Competing interests** None declared.

**Patient consent** Not required.

**Ethics approval** University of Edinburgh, Centre for Population Health Sciences.

**Provenance and peer review** Not commissioned; externally peer reviewed.

**Data sharing statement** The data analysed within this project are administrative data; the Prescribing Information System (PIS) and the Scottish Morbidity Record – 01 Inpatient (SMR01). Anyone wishing to access the data should apply to the electronic Data Research and Innovation Service (eDRIS): www.isdsscotland.org/Products-and-Services/EDRIS/index.asp?Co=Y. Applications to access the data will need to be approved by the Public Benefit and Privacy Panel for Health and Social Care (PBPP): www.informationgovernance.scot.nhs.uk/pbpphsc/. Instructions and guidance on how to apply to access the data can be found here: www.informationgovernance.scot.nhs.uk/pbpphsc/home/for-applicants/.

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
