## [Reviewer comments · BMJ Open]

ARTICLE DETAILS

TITLE (PROVISIONAL)	Interrupted time series evaluation of the impact of abolishing prescription fees in Scotland on hospital admissions and prescribed medicines
AUTHORS	Williams, Andrew James; Henley, W; Frank, John

VERSION 1 – REVIEW

REVIEWER	Sally Wright Liverpool John Moores University, England
REVIEW RETURNED	21-Jan-2018

GENERAL COMMENTS	Overall a well-written and interesting manuscript. Very relevant to policy makers who will no doubt be interested in the impact of abolishing prescription fees. The study was a detailed and involved one. In places throughout the method, more detail would have been beneficial to clarify the rationale for certain decisions, however the inclusion of the original protocol helped, as there appeared to be more detail within this. I appreciate that it is difficult to include all detail within the confines of the manuscript's word limit. Specific comments: Abstract Worth mentioning that a natural experimental approach was taken within the design. Setting – could mention general practice level. Participants – could mention inclusion criteria, why this number of practices were chosen and the total number of practices across Scotland. For clarity, stating early in the abstract that you are using asthma/COPD as the intervention for this study as opposed to all conditions would be informative. Equally, within the results section 'diabetes' and 'young/older patients' were introduced for the first time without any prior explanation in the abstract. It was difficult to ascertain the relevance of the counterfactuals without reading the entire paper. A brief explanation here would be helpful. Could include the statistics to demonstrate the 'marked reductions' mentioned within the results section The conclusion comes across as contradictory to the results – the results state that there were marked step reductions in admissions relating to asthma/COPD, whereas conclusion states no evidence and doesn't refer to this. Perhaps reword to reflect that despite the marked reductions the evidence was not sufficient to determine if universal free prescriptions was effective/ineffective. Introduction Relevant background information – good introduction to study and rationale. Would be interesting to highlight what, if any, research
--

	has been undertaken within Wales and Northern Ireland after they abolished their prescription fees. Method Could elaborate on exactly what general practice level data was available to collect e.g. just the number of items prescribed by each GP practice? Was the data collected monthly (retrospectively) for the entire study period? No specific mention of attempting to avoid bias. In terms of identifying asthma/COPD patients – was it just patients prescribed inhaled corticosteroids? Why not all inhaled therapies – does this not risk excluding a significant proportion of eligible patients? Rationale for this decision would be helpful. Results Include the total number of practices across Scotland i.e. 796 of... Practices Would be interesting to include the data for total numbers of admissions and prescriptions etc. reported for intervention and counterfactual groups. Discussion It may be helpful to state what the important unanswered questions are that need to be followed up with further research. What would the appropriate research methods be to answer these questions for future research?
--	--

REVIEWER	Isidore Sieleunou Research for Development International, Cameroon
REVIEW RETURNED	29-Jan-2018

GENERAL COMMENTS	The paper is well argued and the authors analyse the impact of abolishing prescription fees in Scotland on hospital admissions and prescriptions. There are a few issues related to research ethics that the authors should clarify. What measures was used to protect patient's privacy and confidentiality?
--

REVIEWER	Pauline Norris School of Pharmacy, University of Otago, New Zealand
REVIEW RETURNED	31-Jan-2018

GENERAL COMMENTS	This paper reports on a valiant attempt to, as the authors say, find a needle in a haystack. The implications of patients having to pay for their medicines is an important topic in health policy. The decisions by Wales, Scotland and Northern Ireland to waive prescription charges completely have been watched with interest by many people around the world. Therefore, this study is of international interest and relevance. However, as the authors show, most prescriptions were free throughout the UK before the abolition of prescription charges. This is quite unlike most other countries. Therefore the transition to free prescriptions was not as dramatic as it would be for other countries. This makes it very difficult to evaluate the impact of the change. The authors provide excellent insight into the difficulties of such an evaluation, and are to be commended for their attempts to address the problems with the data, and their balanced assessment of their findings. The discussion section is excellent in
--

	putting the results in context and in discussing wider issues with this data and with big data in general. I am not an expert in statistics or time series analysis, so I recommend that the journal ensure that at least one of the other reviewer(s) is. Minor points: I would suggest using slightly different keywords, such as “co-payment”, “prescription costs”, “out of pocket payments”, rather than “universal free prescriptions”. I think this would allow the study to be found more easily by people looking at the issue of how much people should pay for prescription medicines. Also I think “prescriptions” should be “prescription medicines”. Page 6, lines 16-20. When I read this sentence I was a bit unsure what it meant, and concerned that some of those in the intervention group might really be in the counterfactual. It became clearer later that this was not the case, but the sentence could be revised to improve clarity. I was not familiar with the term “cash” in relation to prescriptions, in New Zealand we say “pick up” a prescription, although that might be equally confusing to readers. Perhaps “get their medicines dispensed”? I am curious about removing the top 1% of prescription data. Could these be bulk supplies for dispensing practices? In the New Zealand data, we also find large dispensings and these are usually for GPs or practices in rural areas who keep some medicines to give to patients when they are unable to access a pharmacy. Why did the authors decide to use gross ingredient cost rather than the cost of the dispensed prescriptions (I assume this includes dispensing and other fees)? This would be more realistic in terms of costs to the government. Line 16, page 18: “including” is a bit unclear. Are the prescriptions purchased using a pre-payment certificate counted as paid for or not?
--	--

REVIEWER	Mario Mazzocchi University of Bologna
REVIEW RETURNED	09-Mar-2018

GENERAL COMMENTS	I am an applied economist, so that my comments will refer to the methods and model specification only. I think the study is interesting and can provide meaningful results. The quasi-experimental (which the author call natural experiment) data are very appropriate for the applications of counterfactual method. I believe the models used in the current version is unnecessarily restrictive and I suggest an alternative specification using a panel difference-in-difference model with practice-level and time fixed effects and clustered standard errors. Results from a model with this specification would be more credible. My comments: - Given the nature of the data, I believe that a mere "interrupted time series" approach is reductive. The data have a panel (cross sectional + longitudinal) dimension, and the implementation of a
--

	difference-in-difference panel model with covariates is straightforward.  - I suggest the authors refer to Jones, Andrew M., et al. "Do public smoking bans have an impact on active smoking? Evidence from the UK." Health economics 24.2 (2015): 175-192. Here the authors consider a panel of households that are exposed to the policy at different times depending on the country they belong to. The model can be applied with no modification to the situations where different units face the policy at different times. The authors can implement the model with the same dummy variables they use in their current model, which will be the interaction term in the diff-in-diff model. - Relate to the above, the model assumes a linear trend (a slope). This is also unnecessarily restrictive, monthly or quarterly time fixed effects could be easily fitted into this specification. With monthly or quarterly fixed effects across the sample, no need to include the seasonal dummies, which reflect the assumption that seasonality is constant over time - ANother limitation of the current interrupted time series specification is that the autors are assuming that the effects of the abolitions/reductions evolve linearly over time. This is a strong assumption, which could be softened by having differential time effects for practices exposed to the policy and those that are not exposed (please note that this will induce collinearity with the fixed effects, so one might need to decide whether to keep the dummies or rely on differential time effects). - Clustering of standard errors is necessary for this type of models. At least at the practice level, better if there is some wider geographical aggregation (e.g. county?) - Why filling with zeroes all missing values? Is there a strong reason to do that? otherwise, I would simply rely on an unbalanced panel data-set
--	---

VERSION 1 – AUTHOR RESPONSE

Response to Reviewers bmjopen-2017-021318

Thank you to all the reviewers for their comments to improve and strengthen the paper. In order to comprehensively demonstrate how we have addressed each reviewer's comments we have added an 'Authors' response' after each comment including references to the location of any changes in the manuscript. We have reviewed the entire manuscript to bring the word count back within the word limit following the changes we made in response to the reviewers comments. All references to locations within the manuscript relate to the version of the manuscript showing the tracked changes.

Reviewer 1: Sally Wright

a) Overall a well-written and interesting manuscript. Very relevant to policy makers who will no doubt be interested in the impact of abolishing prescription fees.

Authors' response: Thank you.

b) The study was a detailed and involved one. In places throughout the method, more detail would have been beneficial to clarify the rationale for certain decisions, however the inclusion of the original protocol helped, as there appeared to be more detail within this. I appreciate that it is difficult to include all detail within the confines of the manuscript's word limit.

Authors' response: Thank you, we are pleased to hear that the protocol clarified any details which were not clear within the paper. We have addressed each of your comments below which hopefully adds further clarity to the paper. We have reviewed the whole paper and a colleague who is not familiar with the study has also reviewed the paper and we have addressed any omissions or areas of confusion that they identified, within the confines of the word limit. Thank you.

Specific comments:

c) Abstract

Worth mentioning that a natural experimental approach was taken within the design.

Authors' response: Added (page 2, Abstract Design section)

d) Setting – could mention general practice level. Authors' response: Added (page 2, Abstract Setting section)

e) Participants – could mention inclusion criteria, why this number of practices were chosen and the total number of practices across Scotland.

Authors' response: We have amended the text, but are limited in the amount of detail which is possible within the word limit of the abstract (page 2, Abstract Participants section).

f) For clarity, stating early in the abstract that you are using asthma/COPD as the intervention for this study as opposed to all conditions would be informative. Equally, within the results section 'diabetes' and 'young/older patients' were introduced for the first time without any prior explanation in the abstract. It was difficult to ascertain the relevance of the

counterfactuals without reading the entire paper. A brief explanation here would be helpful.

Authors' response: Abstract revised (page 2).

g) Could include the statistics to demonstrate the 'marked reductions' mentioned within the results section. The conclusion comes across as contradictory to the results – the results state that there were marked step reductions in admissions relating to asthma/COPD, whereas conclusion states no evidence and doesn't refer to this. Perhaps reword to reflect that despite the marked reductions the evidence was not sufficient to determine if universal free prescriptions was effective/ineffective.

Authors' response: We have revised the text of the Abstract Conclusion but could only make very minimal changes and remain within the word limit. Unfortunately, as it would mean providing quite a few statistics within the abstract to illustrate the text we cannot add the statistics without removing words.

h) Introduction

Relevant background information – good introduction to study and rationale. Would be interesting to highlight what, if any, research has been undertaken within Wales and Northern Ireland after they abolished their prescription fees.

Authors' response: A brief discussion of the research undertaken, following the abolition of prescription fees in Wales and Northern Ireland, has been added (page 6, first paragraph).

i) Method

Could elaborate on exactly what general practice level data was available to collect e.g. just the number of items prescribed by each GP practice?

Authors' response: A list of all the general practice characteristics data within this study dataset has been added, page 10. A much more extensive list of data is likely to be available on practices in Scotland, however, only those variables deemed relevant to the study and unlikely to make any patient identifiable could be included in this study.

j) Was the data collected monthly (retrospectively) for the entire study period?

Authors' response: Yes, the data used are collected for administrative purposes, to reimburse the hospital or pharmacy for the services provided, so there was no connection between the policy and the data collection.

k) No specific mention of attempting to avoid bias.

Authors' response: We acknowledge that, like any dataset there will be biases within the data itself, and potentially, in the way the data are analysed. However, as the data are those primarily collected for financial purposes within a large publicly-funded service, they represent a very low-cost source of data and the 'official' source of data that would be used in reports for politicians and policymakers.

Collecting more accurate data would be very costly and time consuming. Inaccuracies are more likely in the prescriptions data as these originate from many smaller organisations, whereas the hospital admissions data are recorded by clinical coders within the hospital service and then allocated to the patients' General Practice. However, we have no reason to believe that such inaccuracies would have caused bias in our results, as they are likely random in time and space. By fitting practice as a random effect, we have allowed for some variation between practices, and the results appear to suggest that the hospital admissions data are more reliable than the prescriptions data.

l) In terms of identifying asthma/COPD patients – was it just patients prescribed inhaled corticosteroids? Why not all inhaled therapies – does this not risk excluding a significant proportion of eligible patients? Rationale for this decision would be helpful.

Authors' response: It was hypothesised that the policy would have most impact on those requiring long-term maintenance treatments, where the weekly or daily dosage are likely to be relatively stable compared to the reliever medicines. The inhaled corticosteroids were also felt to be most similar in prescribing and usage pattern to diabetes treatments. The patterns of prescribing and usage of asthma reliever therapies will vary much more between individuals, over time and therefore also across practices, making them less suitable for time series analyses.

Results

m) Include the total number of practices across Scotland i.e. 796 of.... Practices Would be interesting to include the data for total numbers of admissions and prescriptions etc. reported for intervention and counterfactual groups.

Authors' response: The figure of 796 reflected the total number of practices within the models of prescriptions metrics. However, as some practices closed and new ones opened during the study period, we have revised this figure to reflect the number of practices in the model in December 2013 (end of the study period) and given the total number of practices registered in Scotland in December 2013. Due to the time element of the study we would either need to report total number of admissions and prescriptions for each year or month, or select one specific year to report on, and we are not sure what these data would add to the paper.

n) Discussion

It may be helpful to state what the important unanswered questions are that need to be

followed up with further research. What would the appropriate research methods be to answer these questions for future research?

Authors' response: Although we have not provided an explicit list of future research questions and possible methods, we feel that these have been mentioned within the discussion, within the confines of the word limit. We have revised the final sentence of the discussion to be less ambiguous.

Reviewer 2: Isidore Sieleunou

a) The paper is well argued and the authors analyse the impact of abolishing prescription fees in Scotland on hospital admissions and prescriptions.

Authors' response: Thank you

b) There are a few issues related to research ethics that the authors should clarify. What measures was used to protect patient's privacy and confidentiality?

Authors' response: The data analysed within this study do not relate to individuals, but have been combined to the level of the General Practice, which was the first step in protecting patient privacy and confidentiality. With a median practice size of around 4,500 and asthma and diabetes being relatively common conditions, and no additional information (e.g. specific age, gender, ethnicity or location) presented in the results or even included in the dataset, re-identification of an individual is not possible. Within the results presented in the manuscript, regression coefficients relating to the whole sample have been presented rather than any results attributable to any individual and finally the whole project was reviewed by the Privacy Advisory Committee (now the Public Benefit and Privacy Panel for Health and Social Care) as a Confidential Data Release, with very low risk. We do not believe that it would be possible to identify an individual from the data included in the manuscript and the data has been handled securely throughout the project, in line with data protection regulations and institutional requirements.

Reviewer 3: Pauline Norris

a) This paper reports on a valiant attempt to, as the authors say, find a needle in a haystack. The implications of patients having to pay for their medicines is an important topic in health policy. The decisions by Wales, Scotland and Northern Ireland to waive prescription charges completely have been watched with interest by many people around the world. Therefore, this study is of international interest and relevance.

Authors' response: Thank you

b) However, as the authors show, most prescriptions were free throughout the UK before the abolition of prescription charges. This is quite unlike most other countries. Therefore, the

transition to free prescriptions was not as dramatic as it would be for other countries. This makes it very difficult to evaluate the impact of the change. The authors provide excellent insight into the difficulties of such an evaluation, and are to be commended for their attempts to address the problems with the data, and their balanced assessment of their findings. The discussion section is excellent in putting the results in context and in discussing wider issues with this data and with big data in general.

Authors' response: Thank you

c) I am not an expert in statistics or time series analysis, so I recommend that the journal ensure that at least one of the other reviewer(s) is.

Authors' response: Please see Reviewer 4's comments below, as the statistical reviewer, and our responses to their comments.

Minor points:

d) I would suggest using slightly different keywords, such as "co-payment", "prescription costs", "out of pocket payments", rather than "universal free prescriptions". I think this would allow the study to be found more easily by people looking at the issue of how much people should pay for prescription medicines. Also I think "prescriptions" should be "prescription medicines".

Authors' response: Thank you for these suggestions we have reviewed our use of terminology around prescription fees and prescriptions throughout the manuscript and tried to ensure the language is internationally relevant. We have swapped the key word 'universal free prescriptions' with the MeSH (Medical Subject Headings) term 'Fees, Pharmaceutical' in the hope that this will capture terms like co-payment. This term was identified as being a key word on a number of the papers cited within the manuscript. Although, in the UK the prescription fee does not vary with the cost of the medicine, we have also added 'co-payment' as a keyword, as you suggest, to make the paper more discoverable.

e) Page 6, lines 16-20. When I read this sentence, I was a bit unsure what it meant, and concerned that some of those in the intervention group might really be in the

counterfactual. It became clearer later that this was not the case, but the sentence could be revised to improve clarity.

Authors' response: Thank you, we have revised this sentence to avoid any confusion. (page 7, Exposure section)

f) I was not familiar with the term "cash" in relation to prescriptions, in New Zealand we say "pick up" a prescription, although that might be equally confusing to readers. Perhaps "get their medicines dispensed"?

Authors' response: Thank you, as with your earlier comment we have revised the terminology to ensure that it can be understood internationally. (pages 7-8, Exposure section)

g) I am curious about removing the top 1% of prescription data. Could these be bulk supplies for dispensing practices? In the New Zealand data, we also find large dispensings and these

are usually for GPs or practices in rural areas who keep some medicines to give to patients when they are unable to access a pharmacy.

Authors' response: Thank you for raising the potential for this phenomenon within the data. We have consulted the pharmacist within NHS Health Scotland who manages the Prescribing Information System (PIS, Stuart McTaggart) who has confirmed that the type of bulk dispensing you describe does not take place in Scotland. During the study period, manual data entry was more common and Stuart reported that keying errors are likely to be responsible for these large dispensings. Currently the system is thought to be 99% accurate so it was appropriate to remove the top 1% of the data. A comment to this effect has been added to the manuscript (page 9, first paragraph).

h) Why did the authors decide to use gross ingredient cost rather than the cost of the dispensed prescriptions (I assume this includes dispensing and other fees)? This would be more realistic in terms of costs to the government.

Authors' response: Again we consulted with Stuart in order to respond to this comment. He has confirmed that gross ingredient cost is used within Scotland for two reasons:
Gross ingredient cost is not affected by discount (clawback) rates which vary between

dispensaries and over time

Pharmacies in Scotland are not paid a fee for each item dispensed but these costs are covered under the service remuneration agreement.

Therefore, analysing gross ingredient cost ensures that the data are not biased or complicated by specific quirks of the funding system, and gives a metric which has a global meaning. A comment to this effect has been added to the manuscript (pages 9-10).

i) Line 16, page 18: "including" is a bit unclear. Are the prescriptions purchased using a pre-payment certificate counted as paid for or not?

Authors' response: We have amended this sentence to avoid any confusion (page 21, paragraph 1).

Reviewer 4: Mario Mazzocchi

a) I am an applied economist, so that my comments will refer to the methods and model specification only.

Authors' response: Thank you; we appreciate receiving your insightful comments on modelling within this complex analysis.

b) I think the study is interesting and can provide meaningful results. The quasi-experimental (which the author call natural experiment) data are very appropriate for the applications of counterfactual method. I believe the models used in the current version is unnecessarily

restrictive and I suggest an alternative specification using a panel difference-in-difference model with practice-level and time fixed effects and clustered standard errors. Results from a model with this specification would be more credible.

Authors' response: Thank you for this suggestion as you discuss it in more detail below we have responded to each specific comment below.

My comments:

c) Given the nature of the data, I believe that a mere "interrupted time series" approach is reductive. The data have a panel (cross sectional + longitudinal) dimension, and the implementation of a difference-in-difference panel model with covariates is straightforward.

Authors' response: Thank you for this suggestion. We did consider taking the difference-in-differences approach, but one of the greatest limitations of this study has been the challenge of identifying an adequate counterfactual group. As we demonstrate within the paper and the sensitivity analysis specifically, neither of our identified counterfactual groups proved to offer an adequate counterfactual and therefore we did not feel that we could pursue the difference-in-differences approach.

d) I suggest the authors refer to Jones, Andrew M., et al. "Do public smoking bans have an impact on active smoking? Evidence from the UK." *Health economics* 24.2 (2015): 175-192. Here the authors consider a panel of households that are exposed to the policy at different times depending on the country they belong to. The model can be applied with no

modification to the situations where different units face the policy at different times. The authors can implement the model with the same dummy variables they use in their current model, which will be the interaction term in the diff-in-diff model.

Authors' response: Thank you for the recommendation of the paper, which we read with interest. Unlike the study undertaken by Jones et al. (2015), we are only studying one country where the policy changed on the same date, nationwide. As discussed above we did attempt to define intra-country counterfactual groups, but with little success. However, we recognise the restriction of modelling time as a predominantly linear effect (adjusted for seasonality) and therefore we have attempted to fit models with dummy variables for each month within the study period. In the absence of a counterfactual, the coefficients estimated for these dummy variables during the reduction and abolition periods are likely to incorporate some element of the 'treatment effect'. Furthermore, Jones, et al. (2015) had 18 time points, where we have over 100, which we suspect prevented the Poisson models of admissions from resolving.

e) Relate to the above, the model assumes a linear trend (a slope). This is also unnecessarily restrictive, monthly or quarterly time fixed effects could be easily fitted into this specification. With monthly or quarterly fixed effects across the sample, no need to include the seasonal dummies, which reflect the assumption that seasonality is constant over time

Authors' response: As discussed above, we have attempted to fit models with monthly fixed effects as you suggest. However, in the absence of a counterfactual group we believe that the fixed effects during the reduction and abolition periods will incorporate some element of the 'intervention effect'. The sensitivity analysis we had undertaken demonstrated some unusual patterns within the data, and therefore – in order to explore the possibility of non-linear time effects – we have fitted the basic models (non-lagged and not stratified by quintiles of the Scottish Index of Multiple Deprivation), adding quadratic terms for time, time from reduction and time from abolition. Much like the other findings, due to the large sample size the coefficients of the quadratic effects were often statistically significant, and more often statistically significant among the counterfactual groups than the intervention group. The direction and significance of the coefficients for fee reduction and abolition – step changes and slope – were barely affected by the addition of these quadratic effects. To illustrate the impact of including quadratic overall time effects, plots showing the linear and quadratic effects for hospital admissions are given below.

Consequently, as the quadratic time effect complicate the interpretation of the results and they does not appear to make a meaningful difference in the findings we would prefer not to complicate the manuscript unnecessarily.

f) - Another limitation of the current interrupted time series specification is that the authors are assuming that the effects of the abolitions/reductions evolve linearly over time. This is a strong assumption, which could be softened by having differential time effects for practices exposed to the policy and those that are not exposed (please note that this will induce

collinearity with the fixed effects, so one might need to decide whether to keep the dummies or rely on differential time effects).

Authors' response: As with your previous comment, we fitted models with added quadratic effects for time since reductions and abolitions began, but keeping overall time as a linear effect. These coefficients are illustrated for hospital admissions below.

Time since reductions began

Reduction effect - Intervention

Reduction effect - Age control

Reduction effect - Condition control

Time since abolition

Abolition effect - Age control

0 5 10 15 20 25 30

Time (months)

Abolition effect - Intervention

Linear predictor
-7
-8
-9
-10
-11

0 5 10 15 20 25 30

Time (months)

Abolition effect - Condition control

Linear predictor
-9.0
-9.5
-10.0
-10.5

0 5 10 15 20 25 30

Time (months)

As with the other finding within the study the most marked impact of the adding the quadratic effects are found in the counterfactual groups. Particularly in terms of the time since abolition, the

quadratic effects made very minimal differences, and therefore as with our previous response we do not feel that these impacts warrant altering the manuscript.

g) - Clustering of standard errors is necessary for this type of models. At least at the practice level, better if there is some wider geographical aggregation (e.g. county?)

Authors' response: The data analysed are at the practice level (e.g. number of admissions and defined daily doses per practice), so the options for clustering would be to cluster by some geographic unit as you suggest. Health services in Scotland are the responsibility of 14 regional Health Boards (listed in Table 2), we cannot find any reason to expect significant clustering at that level. The abolition fee policy was implemented nationwide on the same date, meaning that differences between Health Boards are less relevant, and consequently – in accordance with the recent paper from Abadie, et al. (2017, 'When Should You Adjust Standard Errors for Clustering?' <https://economics.mit.edu/files/13927>) – we do not believe that clustering is appropriate in this case.

The use of practice-level data also means that we don't have data on individual co-variates like age and gender. The potential co-variates available to use, such as practice size and Scottish Index of Multiple Deprivation, have been analysed in the way we felt were most appropriate and informative to the study and research questions.

h) - Why filling with zeroes all missing values? Is there a strong reason to do that? otherwise, I would simply rely on an unbalanced panel data-set

Authors' response: Yes, there is a strong reason for filling in the missing values with zeros. As these data are related to instances when someone from a specific general practice has been admitted to hospital or dispensed a medicine, missing data represent months when no prescriptions of that type or admissions for that condition occurred from that practice. Then refine this to specific age groups and subsequently, with small practice sizes and conditions which should rarely result in hospitalisation, zeros are likely.

VERSION 2 – REVIEW

REVIEWER	Pauline Norris University of Otago, New Zealand
REVIEW RETURNED	02-May-2018

GENERAL COMMENTS	The authors have addressed my concerns, and in my view, the concerns of the other reviewers.
--

REVIEWER	Mario Mazzocchi University of Bologna
REVIEW RETURNED	07-May-2018

GENERAL COMMENTS	It is difficult for me to comment on this revised version, since the authors did not consider my suggestions for further modelling as viable.
---

	Before getting into a detailed responses to the authors' rebuttal, I would like to make it clear my position about this paper: (a) I feel the authors did a good job with their interrupted time series approach; (b) I believe that further attempts at estimating the treatment effect would add credibility to the findings. If the Editor feels that (a) is enough to warrant publication according to the journal standards, I am not insisting for point (b). 1) Given that I am not familiar with the data (limitations) I am ready accept the point that the counterfactual groups identified in Table 1 cannot be used within a Diff-in-Diff framework. I still have some reservations, as I believe that a well-specified model could deal with the limitations of the counterfactual units. As Diff-in-Diff models are meant to deal with non-randomized counterfactuals, it would be worth to give it a try, even if the final decision could be that they are not appropriate. It would be preferable to decide based on the model output/diagnostics. And - more explicitly - why not using age<18 and age>60 as the counterfactual within the diff-in-diff? 2) The authors write "However, we recognise the restriction of modelling time as a predominantly linear effect (adjusted for seasonality) and therefore we have attempted to fit models with dummy variables for each month within the study period. In the absence of a counterfactual, the coefficients estimated for these dummy variables during the reduction and abolition periods are likely to incorporate some element of the 'treatment effect'." I don't think I can agree with this statement. The authors should be confident that their model can distinguish between time-varying trends and the treatment effect. Of course there is a potential collinearity problem between the treatment variable and the quarterly dummies (I would avoid overspecification with monthly variables) within an interrupted time-series framework including an intercept. However, one could also test a model with no intercept, quarterly fixed effects and the intervention dummy, and the model should be able to identify the treatment effect, wouldn't it? The fact that you have over 100 time points is an advantage, not a limitation... 3) While I appreciate testing for a quadratic trend (which is still a strong assumption), I am confused by the authors showing sensitivity by using the counterfactual groups which they dismissed as not adequate for a diff-in-diff approach... 4) The authors also disagree with my request to provide clustered standard errors. In my discipline one would at least expect authors to show how these standard errors change under clustering. I am also unconvinced that what is discussed in Abadie's still unpublished paper points towards not clustering in this study, could the authors elaborate more on this? Also, if quadratic trends are significant, I would expect to see (at least in an Appendix) how the treatment effect estimate is modified under this assumption. Even if the authors believe that quadratic trends are not the best specification, one may want to see how the estimate of the treatment effects is sensitive to the trend specification.
--	---

	5) I accept the justification provided for assigning zeroes to missing values, why not making this justification very explicit in the text?
--	---

VERSION 2 – AUTHOR RESPONSE

Response to Reviewers bmjopen-2017-021318.R1

Thank you to these two reviewers for their time and effort to continue to improve and strengthen the paper. In order to comprehensively demonstrate how we have addressed each reviewer's comments we have added an 'Authors' response' after each comment including references to the location of any changes in the manuscript. All references to locations within the manuscript relate to the version of the manuscript showing the tracked changes.

Reviewer 3: Pauline Norris

- a) The authors have addressed my concerns, and in my view, the concerns of the other reviewers.

Authors' response: Thank you.

Reviewer 4: Mario Mazzocchi

- a) It is difficult for me to comment on this revised version, since the authors did not consider my suggestions for further modelling as viable.

Authors' response: We regret that the reviewer felt this way about our response to their previous comments, and hope that the additional analyses we have now undertaken address their concerns.

- b) Before getting into a detailed responses to the authors' rebuttal, I would like to make it clear my position about this paper: (a) I feel the authors did a good job with their interrupted time series approach; (b) I believe that further attempts at estimating the

treatment effect would add credibility to the findings. If the Editor feels that (a) is enough to warrant publication according to the journal standards, I am not insisting for point (b).

Authors' response: Thank you for your positive comments regarding our use of the interrupted time series approach. We have now undertaken a number of additional analyses which we believe, as you suggested add to the credibility of the findings. These additional analyses are detailed following the relevant comments below and where appropriate in the manuscript.

- c) 1) Given that I am not familiar with the data (limitations) I am ready accept the point that the counterfactual groups identified in Table 1 cannot be used within a Diff-in-Diff framework. I still have some reservations, as I believe that a well-specified model could deal with the limitations of the counterfactual units. As Diff-in-Diff models are meant to deal with non-randomized counterfactuals, it would be worth to give it a try, even if the final decision could be that they are not appropriate. It would be preferable to decide based on the model output/diagnostics.

And - more explicitly - why not using age<18 and age>60 as the counterfactual within the diff-in-diff?

Authors' response: We agree that the implementation of a well specified difference in differences model could help address some of the limitations of the interrupted time series analyses and consequently have attempted to run such models. In line with your comment

(f) below the difference in differences (prior event rate ratio) models accounted for clustering by health board. We found that models which attempted to model the intervention, age-based counterfactual and condition-based counterfactual simultaneously failed, and therefore the intervention condition was compared with the two counterfactual groups individually (Table 1). Interestingly, the models of prescriptions metrics (Defined Daily Doses and Cost) only ran when the intervention condition was compared to the age-based counterfactual, while the admissions model only ran when compared with the condition-based control. These convergence issues appear to reflect model identifiability problems that are related to characteristics of the data because they were specific to particular combinations of outcomes and exposure groups. While it may be possible to address these issues by further simplifying the model structure and by re-scaling of variables, this would make it more difficult to compare the results with the findings of our interrupted time series analyses and we have not pursued this further.

The results of the differences in differences models are consistent with the findings of the interrupted time series analyses. The impact of the reduction and abolition on prescriptions appears to have been greater in the counterfactual than the intervention group. Defined Daily Doses increased across the study period but to a lesser extent among the intervention group compared to the counterfactual, whereas cost of prescribed medications appears to have decreased among the intervention group compared to the counterfactual. However, admissions among the intervention group appear to have made a small reduction compared to the condition-based counterfactual (those aged 18-59 years taking medications for diabetes mellitus), particularly during the reduction period and at the point of fee abolition. However, given the issues faced in running these models and the unusual time trends identified within the Generalised Additive Mixed Models we do not feel that these findings are sufficient to influence the conclusions of the paper. Furthermore, as the findings may lead the reader

to draw conclusions from the study which we don't feel the data can support, we have chosen not to include them in the manuscript.

Table 1 - Results of difference in differences analysis of the impact of reducing and abolishing prescription fees in Scotland on hospital admissions (incidence rate ratios) and prescribed medicines (defined daily doses and cost) – adjusting for clustering with General Practice and Health Board

		Admissions		Admissions 1-month lag		Admissions 2-month lag		Admissions 3-month lag	
		95%		95%		95%			
		IR	CI	IR	CI	IR	CI	IR	95% CI
Intercept		4.16e ⁻³	3.66e ⁻³ to 4.16e ⁻³	3.67e ⁻³	3.67e ⁻³ to 4.17e ⁻³	3.68e ⁻³	3.68e ⁻³ to 4.16e ⁻³	3.67e ⁻³	3.67e ⁻³ to 4.72e ⁻³
Exposure	Counterfactual	(ref)		(ref)		(ref)		(ref)	
	Intervention	2.43	2.35 to 2.51	2.44	2.36 to 2.52	2.44	2.37 to 2.52	2.44	2.36 to 2.52
Monthly Change		1.00	1.00 to 1.00	1.00	1.00 to 1.01	1.00	1.00 to 1.01	1.00	1.00 to 1.01
Seasonality		Omitted for clarity		Omitted for clarity		Omitted for clarity		Omitted for clarity	
Reduction step change		0.95	0.89 to 1.01	0.95	0.89 to 1.02	0.95	0.89 to 1.02	0.93	0.87 to 0.99
	interaction Counterfactual	(ref)		(ref)		(ref)		(ref)	
	Intervention	1.09	1.02 to 1.17	1.06	0.99 to 1.14	1.07	1.00 to 1.15	1.08	1.01 to 1.16
Reduction change in slope		1.00	1.00 to 1.00	1.00	1.00 to 1.00	1.00	1.00 to 1.00	1.00	1.00 to 1.00
	interaction Counterfactual	(ref)		(ref)		(ref)		(ref)	
	Intervention	0.99	0.99 to 1.00	0.99	0.99 to 1.00	0.99	0.99 to 1.00	0.99	0.99 to 1.00
Abolition step change		0.96	0.87 to 1.07	0.97	0.88 to 1.07	0.98	0.89 to 1.08	0.97	0.88 to 1.06
	interaction Counterfactual	(ref)		(ref)		(ref)		(ref)	
	Intervention	0.86	0.80 to 0.92	0.84	0.78 to 0.90	0.84	0.79 to 0.90	0.84	0.79 to 0.90
Abolition change in slope		1.00	1.00 to 1.00	1.00	1.00 to 1.00	1.00	1.00 to 1.00	1.00	1.00 to 1.00
	interaction Counterfactual	(ref)		(ref)		(ref)		(ref)	

Intervention 1.00 0.99 to 1.00 1.00 0.99 to 1.00 1.00 0.99 to 1.00 1.00 0.99 to 1.00

		Defined Daily Doses		Cost	
		DDDs	95% CI	£	95% CI
Intercept		103.88	101.13 to 106.62	76.30	73.38 to 79.23
Exposure	Counterfactual	(ref)		(ref)	
	Intervention	-74.66	-75.16 to -74.17	-61.09	-61.53 to -60.65
Monthly Change		-0.09	-0.12 to -0.07	0.34	0.31 to 0.36
Seasonality		Omitted for clarity		Omitted for clarity	
Reduction step change		6.66	5.78 to 7.54	10.82	10.04 to 11.60
interaction	Counterfactual	(ref)		(ref)	
	Intervention	-5.63	-6.69 to -4.58	-13.39	-14.32 to -12.46
Reduction change in slope		0.69	0.65 to 0.73	0.28	0.24 to 0.31
interaction	Counterfactual	(ref)		(ref)	
	Intervention	-0.59	-0.63 to -0.54	-0.59	-0.63 to -0.55
Abolition step change		40.58	38.97 to 42.18	29.04	27.61 to 30.46
interaction	Counterfactual	(ref)		(ref)	
	Intervention	-31.41	-32.51 to -30.31	-39.00	-39.98 to -38.03
Abolition change in slope		0.40	0.35 to 0.44	-0.05	-0.09 to -0.01
interaction	Counterfactual	(ref)		(ref)	
	Intervention	-0.24	-0.29 to -0.19	-0.23	-0.28 to -0.18

*For the admissions model this is the condition-based counterfactual (those aged 18-59 years taking medications for diabetes mellitus), where for the Defined Daily Doses and Cost models this is the age-based counterfactual (those aged <18 or ≥60 years with asthma or Chronic Obstructive Pulmonary Disease (COPD) taking inhaled corticosteroids).

95% CI; 95% confidence interval, DDDs; Defined daily doses per 100 patients per practice per month, IR; Incidence rate per 100 patients per practice per month, £s; gross ingredient cost of medicines before any discount per 100 patients per practice per month

d) 2) The authors write

"However, we recognise the restriction of modelling time as a predominantly linear effect (adjusted for seasonality) and therefore we have attempted to fit models with dummy variables for each month within the study period. In the absence of a counterfactual, the coefficients estimated for these dummy variables during the reduction and abolition periods are likely to incorporate some element of the

'treatment effect'."

I don't think I can agree with this statement. The authors should be confident that their model can distinguish between time-varying trends and the treatment effect. Of course there is a potential collinearity problem between the treatment variable and the quarterly dummies (I would avoid overspecification with monthly variables) within an interrupted time-series framework including an intercept. However, one could also test a model with no intercept, quarterly fixed effects and the intervention dummy,

and the model should be able to identify the treatment effect, wouldn't it? The fact that you have over 100 time points is an advantage, not a limitation...

Authors' response: We are confident that the new difference in differences models are capable of distinguishing the intervention and secular effects and implemented a fixed effect for Quarter in the models as requested. However, the inclusion of the Quarter variable produced a number of collinearities including with the step change variables for reduction and abolition, even when the 1-3 month lags were being analysed and therefore the intervention and quarter variables were not concurrent. Therefore, as the reduction and abolition step changes are critical variables within the study the Quarter variable has not been included.

- e) 3) While I appreciate testing for a quadratic trend (which is still a strong assumption), I am confused by the authors showing sensitivity by using the counterfactual groups which they dismissed as not adequate for a diff-in-diff approach...

Authors' response: The data analysed within this study originate from the Scottish Morbidity Record and the Prescribing Information System, which each take data from hospitals and pharmacies. Consequently, we feel that if there is an issue apparent in one of the intervention or counterfactual groups, we need to assume that the issue also affects the other groups even if it is not apparent. In running the difference in differences models which in response to your comment (c) above it was found the the models of admissions and prescriptions metrics only converged with one of the counterfactual groups (the age-based counterfactual for the prescription metrics and the condition-based counterfactual for the admissions). Consequently, as it was not possible to observe the difference in differences compared with both counterfactuals we don't feel that conclusions should be drawn from the differences in differences models available. However, this demonstrates the need to

utilise both counterfactual groups as they each reveal different limitations within the data and to the analysis.

- f) 4) The authors also disagree with my request to provide clustered standard errors. In my discipline one would at least expect authors to show how these standard errors change under clustering. I am also unconvinced that what is discussed in Abadie's still unpublished paper points towards not clustering in this study, could the authors

elaborate more on this?

Authors' response: The dataset included information on the Health Board of each General Practice. In Scotland, the regional health boards have responsibility 'for the protection and the improvement of their population's health and for the delivery of frontline healthcare services.'

(<http://www.gov.scot/Topics/Health/NHS-Workforce/NHS-Boards>) Consequently, the health boards have now been used to cluster the standard errors. During the study period there was a change in the number of health boards (in 2006), and therefore the Health Board used is the one from 2008 when the reduction in fees began. The new Supplemental file 4 contains two tables (Table S4.1 and S4.2) that replicate the manuscript Tables 3 and 4 but having accounted for the clustering by Health Board. The estimated effects in Tables S4.1 and S4.2 demonstrate that clustering by Health Board as well as General Practice made very little difference to the findings. The model components which changed most often was the intercept while the other components changed very little. Adjusting the age-based counterfactual model of DDDs for clustering by Health Board caused the model to fail to converge. The manuscript results section has been amended in light of these additional findings (pages 18-19).

- g) Also, if quadratic trends are significant, I would expect to see (at least in an Appendix) how the treatment effect estimate is modified under this assumption. Even if the authors believe that quadratic trends are not the best specification, one may want to see how the estimate of the treatment effects is sensitive to the trend specification.

Authors' response: New Supplemental File 5 provide the estimates from the models we ran for your previous comments, from which the plots were derived (Table S5). As with our previous response, although the inclusion of the quadratic effects alters the estimates, the interpretation remains unchanged.

The plots of the reduction quadratic time effect provided in our previous response included too many months on the x-axis, which may lead to the interpretation that the abolition time effects are additive, however, the variables were specified to be mutually exclusive. Due to the difficulty in interpreting additive quadratic effects, different models were estimated with a quadratic secular time effect and the reduction and abolition effects. The corrected plots are provided in Figure S5 within Supplemental File 5. The manuscript results section has been amended in light of these additional findings (pages 18-19).

- h) 5) I accept the justification provided for assigning zeroes to missing values, why not making this justification very explicit in the text?

Author's response: We are glad that the additional explanation of the justification for this decision is acceptable and have consequently adapted some of this text into the manuscript (page 9, Interrupted Time Series analysis paragraph).

VERSION 3 – REVIEW

REVIEWER	Mario Mazzocchi University of Bologna, Italy
REVIEW RETURNED	01-Aug-2018
GENERAL COMMENTS	I believe the authors have made a commendable effort to test the robustness of their analysis and to document their study - I have no further comments.